# Self-supervised Visual Reinforcement Learning with Object-centric Representations

**Andrii Zadaianchuk**[1,2*]**, Maximilian Seitzer**[1*]**, Georg Martius**[1]
[1] Max Planck Institute for Intelligent Systems, Tübingen, Germany
[2] Department of Computer Science, ETH Zurich
`andrii.zadaianchuk@tuebingen.mpg.de`

## Abstract

Autonomous agents need large repertoires of skills to act reasonably on new tasks that they have not seen before. However, acquiring these skills using only a stream of high-dimensional, unstructured, and unlabeled observations is a tricky challenge for any autonomous agent. Previous methods have used variational autoencoders to encode a scene into a low-dimensional vector that can be used as a goal for an agent to discover new skills. Nevertheless, in compositional/multi-object environments it is difficult to disentangle all the factors of variation into such a fixed-length representation of the whole scene. We propose to use *object-centric representations* as a modular and structured observation space, which is learned with a compositional generative world model. We show that the structure in the representations in combination with *goal-conditioned attention policies* helps the autonomous agent to discover and learn useful skills. These skills can be further combined to address compositional tasks like the manipulation of several different objects.

`https://martius-lab.github.io/SMORL`

## 1 Introduction

Reinforcement learning (RL) includes a promising class of algorithms that have shown capability to solve challenging tasks when those tasks are well specified by suitable reward functions. However, in the real world, people are rarely given a well-defined reward function. Indeed, humans are excellent at setting their own abstract goals and achieving them. Agents that exist persistently in the world should likewise prepare themselves to solve diverse tasks by first constructing plausible goal spaces, setting their own goals within these spaces, and then trying to achieve them. In this way, they can learn about the world around them.

In principle, the goal space for an autonomous agent could be any arbitrary function of the state space. However, when the state space is high-dimensional and unstructured, such as only images, it is desirable to have goal spaces which allow efficient exploration and learning, where the factors of variation in the environment are well disentangled. Recently, unsupervised representation learning has been proposed to learn such goal spaces (Nair et al., 2018; 2019; Pong et al., 2020). All existing methods in this context use variational autoencoders (VAEs) to map observations into a low-dimensional latent space that can later be used for sampling goals and reward shaping.

However, for complex compositional scenes consisting of multiple objects, the inductive bias of VAEs could be harmful. In contrast, representing perceptual observations in terms of entities has been shown to improve data efficiency and transfer performance on a wide range of tasks (Burgess et al., 2019). Recent research has proposed a range of methods for unsupervised scene and video decomposition (Greff et al., 2017; Kosiorek et al., 2018; Burgess et al., 2019; Greff et al., 2019; Jiang et al., 2019; Weis et al., 2020; Locatello et al., 2020). These methods learn object representations and scene decomposition jointly. The majority of them are in part motivated by the fact that the learned representations are useful for downstream tasks such as image classification, object detection, or semantic segmentation. In this work, we show that such learned representations are also beneficial for autonomous control and reinforcement learning.

---

*equal contribution

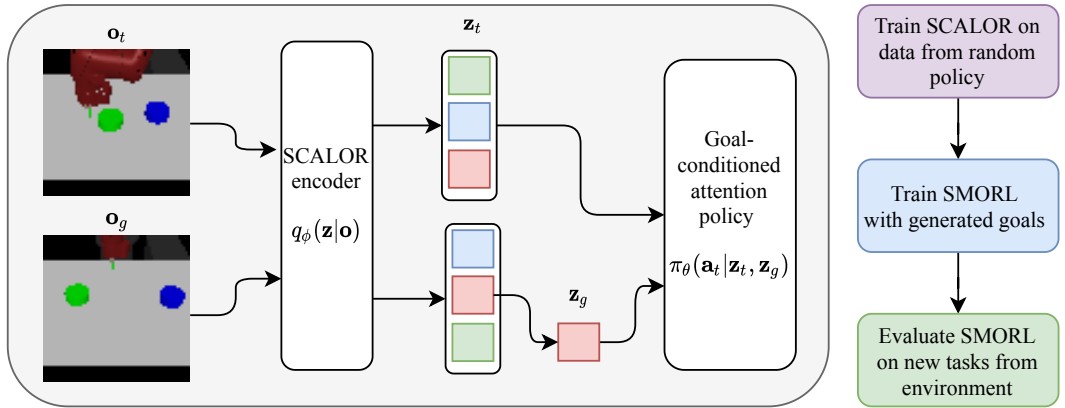

Figure 1: Our proposed SMORL architecture. Representations $\mathbf{z}_t$ are obtained from observations $\mathbf{o}_t$ through the object-centric SCALOR encoder $q_\phi$, and processed by the goal-conditional attention policy $\pi_\theta(\mathbf{a}_t|\mathbf{z}_t, \mathbf{z}_g)$. During training, representations of goals are sampled conditionally on the representations of the first observation $\mathbf{z}_1$. At test time, the agent is provided with an external goal image $\mathbf{o}_g$ that is processed with the same SCALOR encoder to a set of potential goals $\{\mathbf{z}_n\}_{n=1}^N$. After this, the goal $\mathbf{z}_g$ is sequentially chosen from this set. This way, the agent attempts to solve all the discovered sub-tasks one-by-one, not simultaneously.

We propose to combine these *object-centric unsupervised representation* methods that represent the scene as a set of potentially structured vectors with goal-conditional visual RL. In our method (illustrated in Figure 1), dubbed SMORL (for self-supervised multi-object RL), a representation of raw sensory inputs is learned by a compositional latent variable model based on the SCALOR architecture (Jiang et al., 2019). We show that using object-centric representations simplifies the goal space learning. Autonomous agents can use those representations to learn how to achieve different goals with a reward function that utilizes the structure of the learned goal space. Our main contributions are as follows:

- We show that structured object-centric representations learned with generative world models can significantly improve the performance of the self-supervised visual RL agent.
- We develop SMORL, an algorithm that uses learned representations to autonomously discover and learn useful skills in compositional environments with several objects using only images as inputs.
- We show that even with fully disentangled ground-truth representation there is a large benefit from using SMORL in environments with complex compositional tasks such as rearranging many objects.

## 2 RELATED WORK

Our work lies in the intersection of several actively evolving topics: visual reinforcement learning for control and robotics, and self-supervised learning. *Vision-based RL* for robotics is able to efficiently learn a variety of behaviors such as grasping, pushing and navigation (Levine et al., 2016; Pathak et al., 2018; Levine et al., 2018; Kalashnikov et al., 2018) using only images and rewards as input signals. *Self-supervised learning* is a form of unsupervised learning where the data provides the supervision. It was successfully used to learn powerful representations for downstream tasks in natural language processing (Devlin et al., 2018; Radford et al., 2019) and computer vision (He et al., 2019; Chen et al., 2020). In the context of RL, self-supervision refers to the agent constructing its own reward signal and using it to solve self-proposed goals (Baranes & Oudeyer, 2013; Nair et al., 2018; Péré et al., 2018; Hausman et al., 2018; Lynch et al., 2019). This is especially relevant for visual RL, where a reward signal is usually not naturally available. These methods can potentially acquire a diverse repertoire of general-purpose robotic skills that can be reused and combined during test time. Such self-supervised approaches are crucial for scaling learning from narrow single-task learning to more general agents that explore the environment on their own to prepare for solving

many different tasks in the future. Next, we will cover the two most related lines of research in more detail.

**Self-supervised visual RL** (Nair et al., 2018; 2019; Pong et al., 2020; Ghosh et al., 2019; Warde-Farley et al., 2019; Laversanne-Finot et al., 2018) tackles multi-task RL problems from images without any external reward signal. However, all previous methods assume that the environment observation can be encoded into a single vector, e.g. using VAE representations. With multiple objects being present, this assumption may result in object encodings overlapping in the representation, which is known as the binding problem (Greff et al., 2016; 2020). In addition, as the reward is also constructed based on this vector, the agent is incentivized to solve tasks that are incompatible, for instance simultaneously moving all objects to goal positions. In contrast, we suggest to learn object-centric representations and use them for reward shaping. This way, the agent can learn to solve tasks independently and then combine these skills during evaluation.

**Learning object-centric representations in RL** (Watters et al., 2019; van Steenkiste et al., 2019; Veerapaneni et al., 2020; Kipf et al., 2020) has been suggested to approach tasks with combinatorial and compositional elements such as the manipulation of multiple objects. However, the previous work has assumed a fixed, single task and a given reward signal, whereas we are using the learned object-representations to construct a reward signal that helps to learn useful skills that can be used to solve multiple tasks. In addition, these methods use scene-mixture models such as MONET (Burgess et al., 2019) and IODINE (Greff et al., 2019), which do not explicitly contain features like position and scale. These features can be used by the agent for more efficient sampling from the goal space and thus the explicit modeling of these features helps to create additional biases useful for manipulation tasks. However, we expect that other object-centric representations could also be successfully applied as suitable representations for RL tasks.

## 3 BACKGROUND

Our method combines goal-conditional RL with unsupervised object-oriented representation learning for multi-object environments. Before we describe each technique in detail, we briefly state some RL preliminaries. We consider a Markov decision process defined by $(\mathcal{S}, \mathcal{A}, p, r)$, where $\mathcal{S}$ and $\mathcal{A}$ are the continuous state and action spaces, $p \colon \mathcal{S} \times \mathcal{S} \times \mathcal{A} \mapsto [0, \infty)$ is an unknown probability density representing the probability of transitioning to state $\mathbf{s}_{t+1} \in \mathcal{S}$ from state $\mathbf{s}_t \in \mathcal{S}$ given action $\mathbf{a}_t \in \mathcal{A}$, and $r \colon \mathcal{S} \mapsto \mathbb{R}$ is a function computing the reward for reaching state $\mathbf{s}_{t+1}$. The agent's objective is to maximize the expected return $R = \sum_{t=1}^{T} \mathbb{E}_{\mathbf{s}_t \sim \rho_\pi, \mathbf{a}_t \sim \pi, \mathbf{s}_{t+1} \sim p} [r(\mathbf{s}_{t+1})]$ over the horizon $T$, where $\rho_\pi(\mathbf{s}_t)$ is the state marginal distribution induced by the agent's policy $\pi(\mathbf{a}_t | \mathbf{s}_t)$.

### 3.1 GOAL-CONDITIONAL REINFORCEMENT LEARNING

In the standard RL setting described before, the agent only learns to solve a single task, specified by the reward function. If we are interested in an agent that can solve multiple tasks (each with a different reward function) in an environment, we can train the agent on those tasks by telling the agent which distinct task to solve at each time step. But how can we describe a task to the agent? A simple, yet not too restrictive way is to let each task correspond to an environment state the agent has to reach, denoted as the goal state $g$. The task is then given to the agent by conditioning its policy $\pi(a_t \mid s_t, g)$ on the goal $g$, and the agent's objective turns to maximize the expected goal-conditional return:

$$\mathbb{E}_{\mathbf{g} \sim G} \left[ \sum_{t=1}^{T} \mathbb{E}_{\mathbf{s}_t \sim \rho_\pi, \mathbf{a}_t \sim \pi, \mathbf{s}_{t+1} \sim p} [r_{\mathbf{g}}(\mathbf{s}_{t+1})] \right] \qquad (1)$$

where $G$ is some distribution over the space of goals $\mathcal{G} \subseteq \mathcal{S}$ the agent receives for training. The reward function can, for example, be the negative distance of the current state to the goal: $r_{\mathbf{g}}(\mathbf{s}) = -\|\mathbf{s} - \mathbf{g}\|$. Often, we are only interested in reaching a partial state configuration, e.g. moving an object to a target position, and want to avoid using the full environment state as the goal. In this case, we have to provide a mapping $m \colon \mathcal{S} \mapsto \mathcal{G}$ of states to the desired goal space; the mapping is then used to compute the reward function, i.e. $r_{\mathbf{g}}(\mathbf{s}) = -\|m(\mathbf{s}) - \mathbf{g}\|$.

As the reward is computed within the goal space, it is clear that the choice of goal space plays a crucial role in determining the difficulty of the learning task. If the goal space is low-dimensional and structured, e.g. in terms of ground truth positions of objects, rewards provide a meaningful signal towards reaching goals. However, if we only have access to high-dimensional, unstructured observations, e.g. camera images, and we naively choose this space as the goal space, optimization becomes hard as there is little correspondence between the reward and the distance of the underlying world states (Nair et al., 2018).

One option to deal with such difficult observation spaces is to *learn a goal space* in which the RL task becomes easier. For instance, we can try to find a low-dimensional latent space $\mathcal{Z}$ and use it both as the input space to our policy and the space in which we specify goals. If the environment is composed of independent parts that we intend to control separately, intuitively, learning to control is easiest if the latent space is also structured in terms of those independent components. Previous research (Nair et al., 2018; Pong et al., 2020) relied on the disentangling properties of representation learning models such as the $\beta$-VAE (Higgins et al., 2017) for this purpose. However, these models become insufficient when faced with multi-object scenarios due to the increasing combinatorial complexity of the scene, as we show in Sec. 5.2 and in App. A.2. Instead, we use a model explicitly geared towards inferring object-structured representations, which we introduce in the next section.

## 3.2 STRUCTURED REPRESENTATION LEARNING WITH SCALOR

SCALOR (Jiang et al., 2019) is a probabilistic generative world model for learning object-oriented representations of a video or stream of high-dimensional environment observations. SCALOR assumes that the environment observation $\mathbf{o}_t$ at step $t$ is generated by the background latent variable $\mathbf{z}_t^{\text{bg}}$ and the foreground latent variable $\mathbf{z}_t^{\text{fg}}$. The foreground is further factorized into a set of object representations $\mathbf{z}_t^{\text{fg}} = \{\mathbf{z}_{t,n}\}_{n \in \mathcal{O}_t}$, where $\mathcal{O}_t$ is the set of recognised object indices. To combine the information from previous time steps, a propagation-discovery model is used (Kosiorek et al., 2018). In SCALOR, an object is represented by $\mathbf{z}_{t,n} = (z_{t,n}^{\text{pres}}, \mathbf{z}_{t,n}^{\text{where}}, \mathbf{z}_{t,n}^{\text{what}})$. The scalar $z_{t,n}^{\text{pres}}$ defines if the object is present in the scene, whereas the vector $\mathbf{z}_{t,n}^{\text{what}}$ encodes object appearance. The component $\mathbf{z}_{t,n}^{\text{where}}$ is further decomposed into the object's center position $\mathbf{z}_{t,n}^{\text{pos}}$, scale $\mathbf{z}_{t,n}^{\text{scale}}$, and depth $z_{t,n}^{\text{depth}}$. With this, the generative process of SCALOR can be written as:

$$p(\mathbf{o}_{1:T}, \mathbf{z}_{1:T}) = p(\mathbf{z}_1^{\mathcal{D}})p(\mathbf{z}_1^{\text{bg}}) \prod_{t=2}^{T} \underbrace{p(\mathbf{o}_t \mid \mathbf{z}_t)}_{\text{rendering}} \underbrace{p(\mathbf{z}_t^{\text{bg}} \mid \mathbf{z}_{<t}^{\text{bg}}, \mathbf{z}_t^{\text{fg}})}_{\text{background transition}} \underbrace{p(\mathbf{z}_t^{\mathcal{D}} \mid \mathbf{z}_t^{\mathcal{P}})}_{\text{discovery}} \underbrace{p(\mathbf{z}_t^{\mathcal{P}} \mid \mathbf{z}_{<t})}_{\text{propagation}}, \quad (2)$$

where $\mathbf{z}_t = (\mathbf{z}_t^{\text{bg}}, \mathbf{z}_t^{\text{fg}})$, $\mathbf{z}_t^{\mathcal{D}}$ contains latent variables of objects discovered in the present step, and $\mathbf{z}_t^{\mathcal{P}}$ contains latent variables of objects propagated from the previous step. Due to the intractability of the true posterior distribution $p(\mathbf{z}_{1:T}|\mathbf{o}_{1:T})$, SCALOR is trained using variational inference with the following posterior approximation:

$$q(\mathbf{z}_{1:T} \mid \mathbf{o}_{1:T}) = \prod_{t=1}^{T} q(\mathbf{z}_t \mid \mathbf{z}_{<t}, \mathbf{o}_{\leq t}) = \prod_{t=1}^{T} q(\mathbf{z}_t^{\text{bg}} \mid \mathbf{z}_t^{\text{fg}}, \mathbf{o}_t) \, q(\mathbf{z}_t^{\mathcal{D}} \mid \mathbf{z}_t^{\mathcal{P}}, \mathbf{o}_{\leq t}) \, q(\mathbf{z}_t^{\mathcal{P}} \mid \mathbf{z}_{<t}, \mathbf{o}_{\leq t}), \quad (3)$$

by maximizing the following evidence lower bound $\mathcal{L}(\theta, \phi) =$

$$\sum_{t=1}^{T} \mathbb{E}_{q_\phi(\mathbf{z}_{<t}|\mathbf{o}_{<t})} \Big[ \mathbb{E}_{q_\phi(\mathbf{z}_t|\mathbf{z}_{<t}, \mathbf{o}_{\leq t})} \big[ \log p_\theta(\mathbf{o}_t \mid \mathbf{z}_t) \big] - D_{\text{KL}} \big[ q_\phi(\mathbf{z}_t \mid \mathbf{z}_{<t}, \mathbf{o}_{\leq t}) \parallel p_\theta(\mathbf{z}_t \mid \mathbf{z}_{<t}) \big] \Big], \quad (4)$$

where $D_{\text{KL}}$ denotes the Kullback-Leibler divergence. As we are using SCALOR in an active setting, we additionally condition the next step posterior predictions on the actions $\mathbf{a}_t$ taken by the agent. For more details and hyperparameters used to train SCALOR, we refer to App. D.3. In the next section, we describe how the structured representations learned by SCALOR can be used in downstream RL tasks such as goal-conditional visual RL.

# 4 SELF-SUPERVISED MULTI-OBJECT REINFORCEMENT LEARNING

Learning from flexible representations obtained from unsupervised scene decomposition methods such as SCALOR creates several challenges for RL agents. In particular, these representations consist of sets of vectors, whereas standard policy architectures assume fixed-length state vectors as input. We propose to use a *goal-conditioned attention policy* that can handle sets as inputs and flexibly learns to attend to those parts of the representation needed to achieve the goal at hand.

In the setting we consider, the agent is not given *any reward signal or goals from the environment* at the training stage. Thus, to discover useful skills that can be used during evaluation tasks, the agent needs to rely on *self-supervision* in the form of an internally constructed reward signal and self-proposed goals. Previous VAE-based methods used latent distances to the goal state as the reward signal. However, for compositional goals, this means that the agent needs to master the simultaneous manipulation of all objects. In our experiments in Sec. 5.1, we show that even with fully disentangled, ground-truth representations of the scene, this is a challenging setting for state-of-the-art model-free RL agents. Instead, we propose to use the discovered structure of the learned goal and state spaces twofold: the structure within each representation, namely object position and appearance, to construct a reward signal, and the set-based structure between representations to construct sub-goals that correspond to manipulating individual objects.

## 4.1 POLICY WITH GOAL-CONDITIONED ATTENTION

We use the multi-head attention mechanism (Vaswani et al., 2017) as the first stage of our policy $\pi_\theta$ to deal with the challenge of set-based input representations. As the policy needs to flexibly vary its behavior based on the goal at hand, it appears sensible to steer the attention using a goal-dependent query $Q(\mathbf{z}_g) = \mathbf{z}_g W^q$. Each object is allowed to match with the query via an object-dependent key $K(\mathbf{z}_t) = \mathbf{z}_t W^k$ and contribute to the attention's output through the value $V(\mathbf{z}_t) = \mathbf{z}_t W^v$, which is weighted by the similarity between $Q(\mathbf{z}_g)$ and $K(\mathbf{z}_t)$. As inputs, we concatenate the representations for object $n$ to vectors $\mathbf{z}_{t,n} = [\mathbf{z}_{t,n}^{\text{what}}; \mathbf{z}_{t,n}^{\text{where}}; z_{t,n}^{\text{depth}}]$, and similarly the goal representation to $\mathbf{z}_g = [\mathbf{z}_g^{\text{what}}; \mathbf{z}_g^{\text{where}}; z_g^{\text{depth}}]$. The attention head $A_k$ is computed as

$$A_k = \text{softmax}\left(\frac{\mathbf{z}_g W^q (Z_t W^k)^T}{\sqrt{d_e}}\right) Z_t W^v, \tag{5}$$

where $Z_t$ is a packed matrix of all $\mathbf{z}_{t,n}$'s, $W^q$, $W^k$, $W^v$ constitute learned linear transformations and $d_e$ is the common key, value and query dimensionality. The final attention output $A$ is a concatenation of all the attention heads $A = [A_1; \ldots; A_K]$. In general, we expect it to be beneficial for the policy to not only attend to entities conditional on the goal; we thus let some heads attend based on a set of input independent, learned queries, which are not conditioned on the goal. We go into more details about the attention mechanism in App. D.1 and ablate the impact of different choices in App. B.

The second stage of our policy is a fully-connected neural network $f$ that takes as inputs $A$ and the goal representation $\mathbf{z}_g$ and outputs an action $\mathbf{a}_t$. The full policy $\pi_\theta$ can thus be described by

$$\pi_\theta\left(\{\mathbf{z}_{t,n}\}_{n \in \mathcal{O}_t}, \mathbf{z}_g\right) = f(A, \mathbf{z}_g). \tag{6}$$

## 4.2 SELF-SUPERVISED TRAINING

In principle, our policy can be trained with any goal-conditional model-free RL algorithm. For our experiments, we picked soft-actor critic (SAC) (Haarnoja et al., 2018b) as a state-of-the-art method for continuous action spaces, using hindsight experience replay (HER) (Andrychowicz et al., 2017) as a standard way to improve sample-efficiency in the goal-conditional setting.

The training algorithm is summarized in Alg. 1. We first train SCALOR on data collected from a random policy and fit a distribution $p(\mathbf{z}^{\text{where}})$ to representations $\mathbf{z}^{\text{where}}$ of collected data. Each rollout, we generate a new goal for the agent by picking a random $\mathbf{z}^{\text{what}}$ from the initial observation $\mathbf{z}_1$ and sampling a new $\mathbf{z}^{\text{where}}$ from the fitted distribution $p(\mathbf{z}^{\text{where}})$. The policy is then rolled out using this goal. During off-policy training, we are relabeling goals with HER, and, similar to RIG (Nair et al., 2018), also with "imagined goals" produced in the same way as the rollout goals.

**Algorithm 1** SMORL: Self-Supervised Multi-Object RL (Training)

---

**Require:** SCALOR encoder $q_\phi$, goal-conditional policy $\pi_\theta$, goal-conditional SAC trainer, number of training
    episodes $K$.
1: Train SCALOR on sequences uniformly sampled from $\mathcal{D}$ using loss described in Eq. 4.
2: Fit prior $p(\mathbf{z}^{\text{where}} \mid \mathbf{z}^{\text{what}})$ to the latent encodings of observations.
3: **for** $n = 1, ..., K$ episodes **do**
4:    Sample goal $\mathbf{z}_g = \left( \hat{\mathbf{z}}_g^{\text{where}}, \mathbf{z}_g^{\text{what}} \right)$.
5:    Collect episode data with policy $\pi_\theta(\mathbf{a}_t \mid \mathbf{z}_t, \mathbf{z}_g)$ and SCALOR representations of observations $q_\phi(\mathbf{z}_t \mid$
       $\mathbf{z}_{<t}, \mathbf{o}_{\leq t})$.
6:    Store transitions $(\mathbf{z}_t, \mathbf{a}_t, \mathbf{z}_{t+1}, \mathbf{z}_g)$ into replay buffer $\mathcal{R}$.
7:    Sample transitions from replay buffer $(\mathbf{z}, \mathbf{a}, \mathbf{z}', \mathbf{z}_g) \sim \mathcal{R}$.
8:    Relabel $\mathbf{z}_g^{\text{where}}$ goal components to a combination of future states and $p(\mathbf{z}^{\text{where}} \mid \mathbf{z}^{\text{what}})$.
9:    Compute matching reward signal $R = r(\mathbf{z}', \mathbf{z}_g)$.
10:    Update policy $\pi_\theta(\mathbf{a}_t \mid \mathbf{z}_t, \mathbf{z}_g)$ using $R$ with SAC trainer.
11: **end for**

---

We also refer to Alg. 2 in App. D.2 for a more detailed description of the algorithm.

A challenge with compositional representations is how to measure the progress of the agent towards achieving the chosen goal. As the goal always corresponds to a single object, we have to extract the state of this object in the current observation in order to compute a reward. One way is to rely on the tracking of objects, as was shown possible e.g. by SCALOR (Jiang et al., 2019). However, as the agent learns, we noticed that it would discover some flaws of the tracking and exploit them to get a maximal reward that is not connected with environment changes, but rather with internal vision and tracking flaws (details in App. E).

We follow an alternative approach, namely to use the $\mathbf{z}^{\text{what}}$ component of discovered objects and match them with the current goal representation $\mathbf{z}_g^{\text{what}}$. As the $\mathbf{z}^{\text{what}}$ space encodes the appearance of objects, two detections corresponding to the same object should be close in this space (we verify that this hypothesis holds in App. A.1). Thus, it is easy to find the object corresponding to the current goal object using the distance $\min_k ||\mathbf{z}_k^{\text{what}} - \mathbf{z}_g^{\text{what}}||$. In case of failure to discover a close representation, i.e. when all $\mathbf{z}_k^{\text{what}}$ have a distance larger than some threshold $\alpha$ to the goal representation $\mathbf{z}_g^{\text{what}}$, we use a fixed negative reward $r_{\text{no\_goal}}$ to incentivise the agent to avoid this situation.

Our reward signal is thus

$$
r(\mathbf{z}, \mathbf{z}_g) = \begin{cases} -||\mathbf{z}_{\hat{k}}^{\text{where}} - \mathbf{z}_g^{\text{where}}|| & \text{if } \min_k ||\mathbf{z}_k^{\text{what}} - \mathbf{z}_g^{\text{what}}|| < \alpha, \\ r_{\text{no\_goal}} & \text{otherwise,} \end{cases} \tag{7}
$$

where $\hat{k} = \arg \min_k ||\mathbf{z}_k^{\text{what}} - \mathbf{z}_g^{\text{what}}||$.

### 4.3 Composing Independent Sub-Goals during Evaluation

At evaluation time, the agent receives a goal image from the environment showing the state to achieve. The goal image is processed by SCALOR to yield a set of goal vectors. For our experiments, we assume that these sub-goals are independent of each other and that the agent can thus sequentially achieve them by cycling through them until all of them are solved. The evaluation algorithm is summarized in Alg. 3, with more details added in App. D.2.

## 5 Experiments

We have done computational experiments to address the following questions:

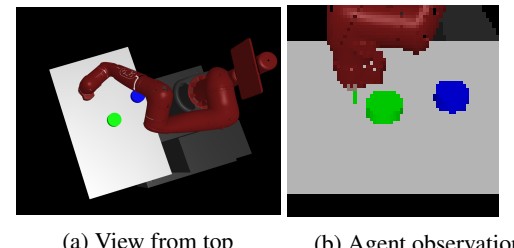

(a) View from top      (b) Agent observation

Figure 2: Multi-Object Visual Push and Rearrange environments with 2 objects and a Sawyer robotic arm.

- How well does our method scale to challenging tasks with a large number of objects in case when ground-truth representations are provided?

- How does our method perform compared to prior visual goal-conditioned RL methods on image-based, multi-object continuous control tasks?

- How suitable are the representations learned by the compositional generative world model for discovering and solving RL tasks?

To answer these questions, we constructed the *Multi-Object Visual Push* and *Multi-Object Visual Rearrange* environments. Both environments are based on MuJoCo (Todorov et al., 2012) and the Multiworld package for image-based continuous control tasks introduced by Nair et al. (2018), and contain a 7-dof Sawyer arm where the agent needs to be controlled to manipulate a variable number of small picks on a table. In the first environment, the objects are located on fixed positions in front of the robot arm that the arm must push to random target positions. We included this environment as it largely corresponds to the *Visual Pusher* environments of Nair et al. (2018). In the second environment, the task is to rearrange the objects from random starting positions to random target positions. This task is more challenging for RL algorithms due to the randomness of initial object positions. For both environments, we measure the performance of the algorithms as the average distance of all pucks to their goal positions on the last step of the episode. Our code, as well as the multi-objects environments will be made public after the paper publication.

### 5.1 SMORL with ground-truth (GT) state representation

We first compared SMORL with ground-truth representation with Soft Actor-Critic (SAC) (Haarnoja et al., 2018a) with Hindsight Experience Replay (HER) relabeling (Andrychowicz et al., 2017) that takes an unstructured vector of all objects coordinates as input. We are using a one-hot encoding for object identities $\mathbf{z}^{\text{what}}$ and object and arm coordinates as $\mathbf{z}^{\text{where}}$ components. With such a representation, the matching task becomes trivial, so our main focus in this experiment is on the benefits of the goal-conditioned attention policy and the sequential solving of independent sub-tasks. We show the results in Fig. 3. While for 2 objects, SAC+HER is performing similarly, for 3 and 4 objects, SAC+HER fails to rearrange any of the objects. In contrast, SMORL equipped with ground-truth representation is still able to rearrange 3 and 4 objects, and it can solve the more simple sub-tasks of moving each object independently. This shows that provided with good representations, SMORL can use them for constructing useful sub-tasks and learn how to solve them.

### 5.2 Visual RL Methods Comparison

We compare the performance of our algorithm with two other self-supervised, multi-task visual RL algorithms on our two environments, with one and two objects. The first one, RIG (Nair et al., 2019), uses the VAE latent space to sample goals and to estimate the reward signal. The second one, Skew-Fit (Pong et al., 2020), also uses the VAE latent space, however, is additionally biased on rare observations that were not modeled well by the VAE on previously collected data. In terms of computational complexity, both our method and RIG need to train a generative model before RL training. We note that training SCALOR is more costly than training RIG's VAE due to the sequence processing utilized by SCALOR. However, once trained, SCALOR only adds little overhead compared to RIG's VAE during RL training, and compared to Skew-Fit, our method is still faster to train as Skew-Fit needs to continuously retrain its VAE.

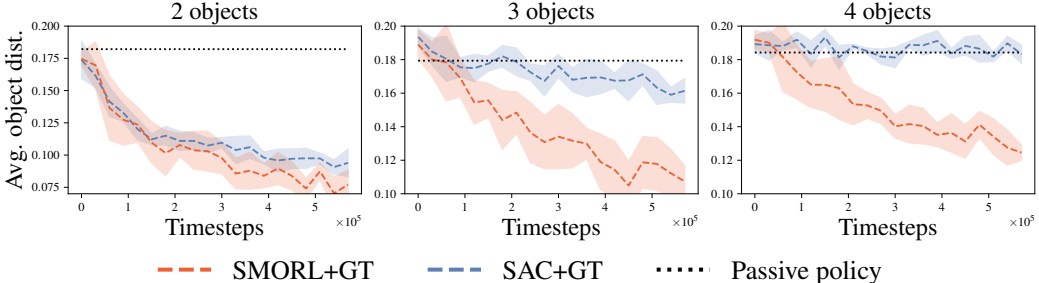

Figure 3: Average distance of objects to goal positions, comparing SMORL using ground truth representations to SAC with ground truth representations in the Rearrange environment with different number of objects. SAC struggles to improve performance when the combinatorial complexity of the scene rises. The dotted line indicates the performance of a *passive policy* that performs no movements. Results averaged over 5 random seeds, shaded regions indicate one standard deviation.

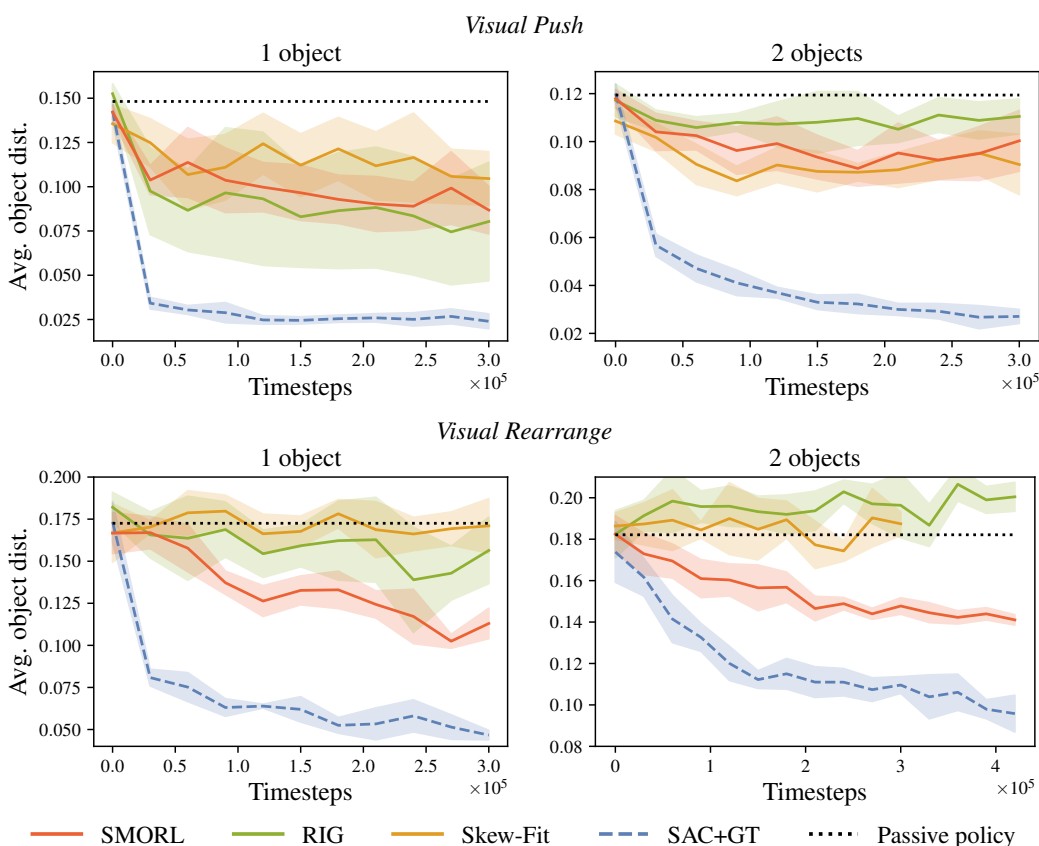

Figure 4: Average distance of objects to goal positions, comparing SMORL to Visual RL Baselines. In addition to the baselines, we show SAC's performance with ground truth representations. Results averaged over 5 random seeds, shaded regions indicate one standard deviation.

We show the results in Fig. 4. For the simpler *Multi-Object Visual Push* environment, the performance of SMORL is comparable to the best performing baseline, while for the more challenging *Multi-Object Visual Rearrange* environment, SMORL is significantly better then both RIG and Skew-Fit. This shows that learning of object-oriented representations brings benefits for goal sampling and self-supervised learning of useful skills. However, our method is still significantly worse than SAC with ground-truth representations. We hypothesize that one reason for this could be that SCALOR right now does not properly deal with occluded objects, which makes the environment

partially observable from the point of view of the agent. On top of this, we suspect noise in the representations, misdetections and an imperfect matching signal to slow down training and ultimately hurt performance. Thus, we expect that adding recurrence to the policy or improving SCALOR itself could help close the gap to an agent with perfect information.

### 5.3 Out-of-Distribution Generalization for different number of objects

One important advantage of structured policies is that they could potentially still be applicable for observations that are from different, but related distributions. Standard visual RL algorithms were shown to be sensitive to small changes unrelated to the current task (Higgins et al., 2018). To see how our algorithm can generalize to a changing environment, we tested our SMORL agent trained on observations of the Rearrange environment with 2 objects on the environment with 1 object. As can be seen from Fig. 5, the performance of such an agent increases during training up to a performance comparable to a SMORL agent that was trained on the 1 object environment.

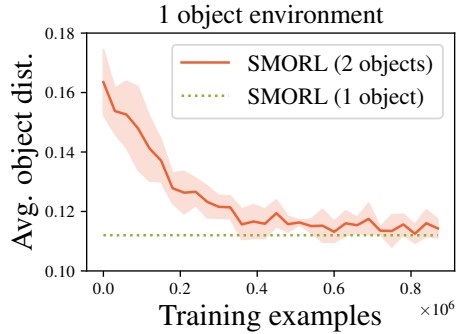

Figure 5: Out-of-distribution generalization of SMORL agent training on *Visual Rearrange* with two objects and being tested with one object. Green line shows final performance when training with one object.

## 6 Conclusion and Future Work

In this work, we have shown that discovering structure in the observations of the environment with a compositional generative world models and using it for controlling different parts of the environment is crucial for solving tasks in compositional environments. Learning to manipulate different parts of object-centric representations is a powerful way to acquire useful skills such as object manipulation. Our SMORL agent learns how to control different entities in the environment and can then combine the learned skills to achieve more complex compositional goals such as rearranging several objects using only the final image of the arrangement.

Given the results presented so far, there are a number of interesting directions to take this work. First, one can combine learned sub-tasks with a planning algorithm to achieve a particular goal. Currently, the agent is simply sequentially cycling through all discovered sub-tasks, so we expect that a more complex planning algorithm as e.g. described by Nasiriany et al. (2019) could allow solving more challenging tasks and improve the overall performance of the policy. To this end, considering interactions between objects in the manner of Fetaya et al. (2018) or Kipf et al. (2020) could help to lift the assumption of independence of sub-tasks. Second, prioritizing certain sub-tasks during learning, similar to Blaes et al. (2019), could accelerate the training of the agent. Finally, an active training of SCALOR to combine the object-oriented bias of SCALOR with a bias towards independently controllable objects (Thomas et al., 2018) is an interesting direction for future research.

### Acknowledgements

The authors thank the International Max Planck Research School for Intelligent Systems (IMPRS-IS) for supporting Maximilian Seitzer. Andrii Zadaianchuk is supported by the Max Planck ETH Center for Learning Systems. We acknowledge the support from the German Federal Ministry of Education and Research (BMBF) through the Tübingen AI Center (FKZ: 01IS18039B).

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

APPENDIX

# A  ANALYSIS OF REPRESENTATIONS LEARNED BY SCALOR

## A.1  CLUSTERING OF $\mathbf{z}^{\text{WHAT}}$ COMPONENTS

In this section, we analyze the representations learned by SCALOR. First, we looked at how well different detections of the same object cluster together in the $\mathbf{z}^{\text{what}}$ space SCALOR learns. This is important in order to find out whether we can use distances in $\mathbf{z}^{\text{what}}$ space to match corresponding objects which is necessary to compute rewards for the agent (see Sec. 4.2). A well separated $\mathbf{z}^{\text{what}}$ space also indicates the usefulness of SCALOR's representations for other potential downstream tasks such as classification. In Fig. 6, we plot the first and second principal component of points in $\mathbf{z}^{\text{what}}$ space, and color each point according to the mean pixel value of the foreground object in the crop detected by SCALOR. As one can see, the three objects (green, blue, and red points) and the robotic arm (darker red points) are quite well separated, with relatively low intra cluster variance. The robotic arm cluster shows larger variance as it is observed in more different poses than the objects. There is also a small cluster of misdetections (center top), with the gray color of the table. Overall, this shows that the $\mathbf{z}^{\text{what}}$ space is well suited for the purpose of matching.

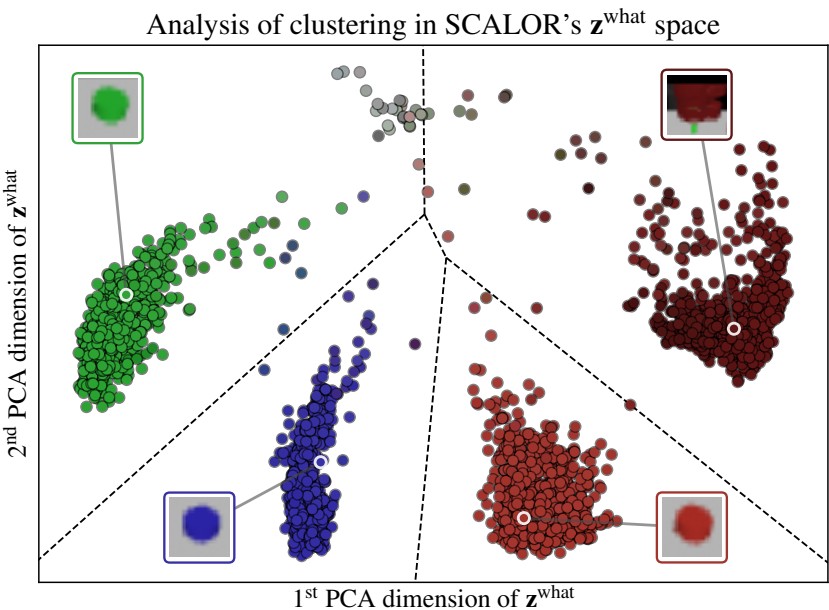

Figure 6: First and second PCA dimension of $\mathbf{z}^{\text{what}}$ space of SCALOR trained on *Visual Rearrange* with 3 objects. The plot shows 3000 random $\mathbf{z}^{\text{what}}$ points collected from a random policy. Each point is colored as the mean of the foreground pixels on the crop detected by SCALOR. For each cluster, the highlighted point shows an example crop. Dashed lines indicate the Voronoi partitions according to cluster centers found by running k-means clustering. Figure is best viewed on screen.

## A.2  DISENTANGLEMENT ANALYSIS OF REPRESENTATIONS LEARNED BY SCALOR AND VAE

After seeing that SCALOR representation can be successfully used for object classification, we further examined the quality of the object location information learned by SCALOR by evaluating how disentangled they are. For this, we computed Mutual Information Gap (MIG) (Chen et al., 2018) scores for SCALOR and VAE components. As SCALOR representations are *unordered sets of vectors*, we used the clusters obtained from the cluster analysis (see App. A.1) to produce a vector $\mathbf{z}_{\text{vec}}^{\text{where}}$ that has consistent dimension ordering by matching $\mathbf{z}^{\text{what}}$ components to clusters. In the case of an object not being recognized in an image, we imputed zeros values to its part in the vector $\mathbf{z}_{\text{vec}}^{\text{where}}$.

We estimated MIG by adapting the `disentanglement_lib` (Locatello et al., 2019), with an additional discretization of the continuous ground truth factors in the same way the continuous latent space is discretized.

The results in Fig. 7 show that SCALOR's $\mathbf{z}^{\text{where}}$ components are more disentangled and thus are better suited for the construction of independent RL sub-tasks. In addition, it can be seen that the VAE disentanglement score is quite low, potentially because different factors of variation (object coordinates) have the same variance and thus could be more difficult to disentangle (Rolinek et al., 2019).

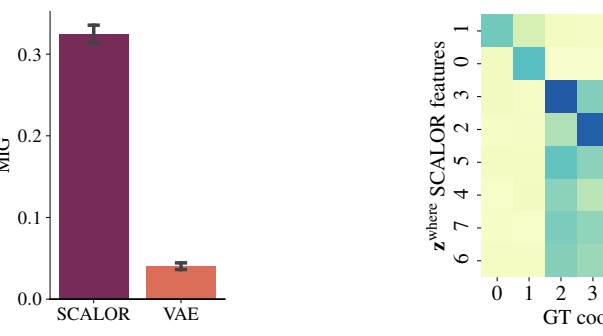

(a) MIG score (higher is better).      (b) Mutual information matrix for SCALOR representations.

Figure 7: Comparison of VAE and SCALOR representations. *(a)* shows MIG scores of VAE and SCALOR representations on data obtained from running a random policy in the *Visual Rearrange* environment with 3 objects (with whisker showing the standard deviation over 5 runs), *(b)* shows the mutual information matrix for SCALOR representations on the same data.

## A.3 SCALOR TRAJECTORY TRAVERSALS

One of the ways to evaluate the quality of learned representations is to show how it reconstructs the scene. To this end, Fig. 8 shows some example environment traversals and how SCALOR processes them. SCALOR is not only able to reconstruct the final image, but in addition is also able to locate objects and produce accurate segmentation masks for each object.

## B ABLATION ANALYSIS OF GOAL-CONDITIONED ATTENTION POLICY

To understand how important the contribution of the goal-conditioned attention policy is to the performance and the generalization properties of our architecture, we have compared it with several other options for processing the set of SCALOR representations. In particular, we test two more variants of our attention mechanism: one where we use only *goal-conditional* attention heads, and one where we use only *goal-unconditional* heads with learned, input-independent queries. We hypothesize that using only goal-conditional heads reduces the ability of the policy to easily concentrate on parts of the environment that are globally relevant for all tasks. Using only goal-unconditional heads with learned queries should still allow the policy to learn to order the input representations and produce a consistent fixed length vector; however, it removes the ability to flexibly select parts of the inputs based on the task at hand. Finally, we also implemented the DeepSet method (Zaheer et al., 2017) as an alternative approach to process inputs of sets of vector representation. In our case, we instantiate DeepSets by transforming each component embedding with a one hidden layer MLP with ReLU activation to feature vectors of dimensionality 128 and then summing up these vectors.

The results in Fig. 9 show that both types of attention heads are necessary to achieve the best results, with the goal-conditional heads having a larger impact on the final performance. Without the goal-conditional heads, the SMORL algorithm performs significantly worse. In addition, we observe that SMORL with DeepSets can also perform competitively on the two objects tasks, however, it is significantly worse on the out-of-distribution task with one object.

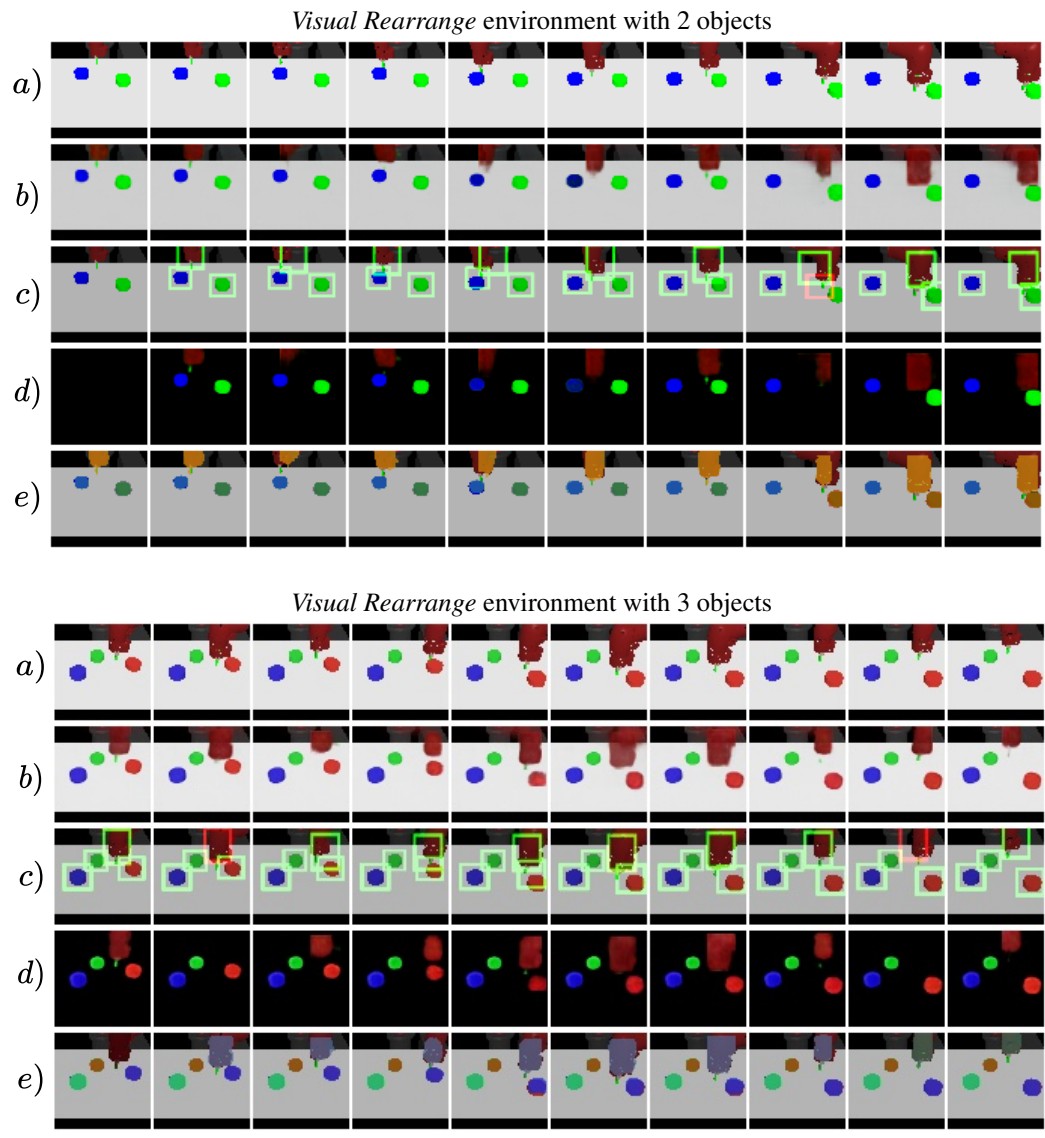

Figure 8: Reconstructions of scene observations using learned SCALOR representation and decoder. Rows are $a$) original images (green boxes for recognized objects, red boxes for non-propagated objects), $b$) full reconstructions, $c$) bounding boxes of recognized objects produced using $\mathbf{z}^{\text{where}}$, $d$) foreground object reconstructions, $e$) segmentation masks of objects generated by SCALOR.

## C  LONGER TRAINING FOR VISUAL REARRANGE WITH TWO OBJECTS

For the challenging *Visual Rearrange* environment with 2 objects, we trained a SMORL agent for twice as long as in the main plot in Fig. 4 to better understand the final convergence performance (see Fig. 10). Whereas the RIG baseline still shows no signs of progress after one million timesteps, our SMORL agent is continuing to improve performance. This result hints at that with even more training steps, SMORL might eventually reach the performance of a SAC agent that has privileged information of the ground-truth state.

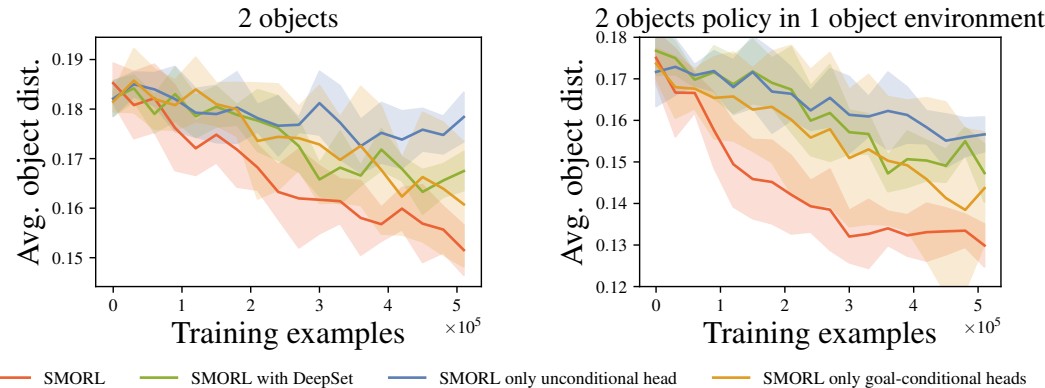

Figure 9: Ablation study of goal-conditioned attention policy on *Visual Rearrange* with two objects (left) and out-of-distribution testing on *Visual Rearrange* with one object (right). We compare variants of the attention policy with only goal-conditional and only goal-unconditional attention heads, plus an alternative approach to aggregate sets of vector representations in the form of DeepSets (Zaheer et al., 2017). Our results demonstrate that both types of attention heads are necessary to achieve the best results.

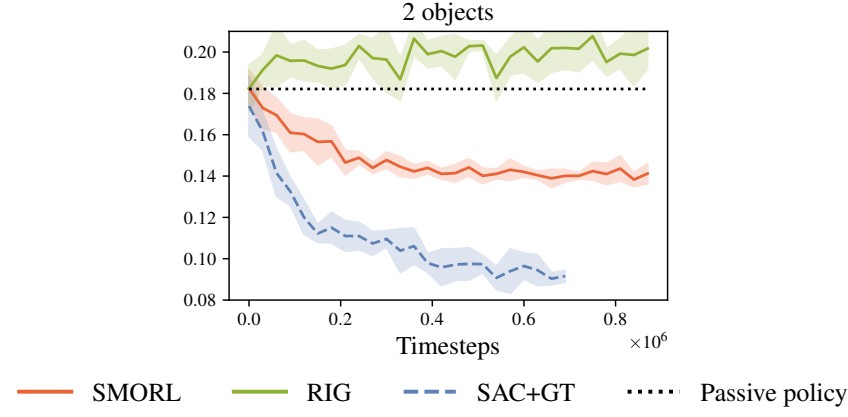

Figure 10: Performance of a SMORL agent trained for $10^6$ timesteps on *Visual Rearrange* with 2 objects.

## D  IMPLEMENTATION DETAILS

### D.1  DETAILS OF ATTENTION MECHANISM

As discussed in Sec. 4.1, the policy should be able to select from the set of input representations based on the goal it needs to solve. We implement this by running attention with a goal-dependent query $Q(\mathbf{z}_g) = \mathbf{z}_g W^q$. However, there might be some parts of the input state that are always relevant to the policy, regardless of the current goal. For example, in our experiments, we expect the state of the robotic arm to always be important, as it is needed to manipulate objects. To simplify the extraction of this information for the policy, we optionally add $M$ learned, input-independent, goal-unconditional queries $Q(P^q)$ to the goal-dependent query. $P^q \in \mathbb{R}^{M \times d_e}$ is simply a matrix of parameters that is trained via backpropagation; we initialize $P^q$ by sampling from $\mathcal{N}(0, 0.02)$. Furthermore, we use two separate sets of attention heads to process the goal-conditional and -unconditional queries, i.e. each set of attention head has its own set of projections $W^q, W^v, W^k$. The output of both sets of attention heads is simply concatenated before feeding it to the next stage of the policy.

We use Pytorch's (Paszke et al., 2019) `torch.nn.MultiheadAttention` module to implement the attention mechanism. In practice, we use two separate instantiations of these modules to implement

goal-conditional and goal-unconditional heads. In accordance to the original transformer attention formulation (Vaswani et al., 2017), this module also includes a linear transformation that mixes the outputs of the different heads together. As we think this transformation is not strictly necessary, we have omitted it for notational clarity. Moreover, note that we also linearly embed the policy inputs $\mathbf{z}^g$ and $\mathbf{z}_{t,n}$'s into a common space of dimensionality $d_e$ before processing them further, which we have found to slightly improve performance.

## D.2 FULL SMORL TRAINING AND EVALUATION ALGORITHMS

We display a fully detailed version of the training algorithm in Alg. 2. In addition, Alg. 3 shows how we apply SMORL during evaluation. For evaluation, the agent receives a goal image to achieve from the environment. After processing this image into latent representations with SCALOR, the agent picks one of the recognized objects as its sub-goal and attempts to achieve it for a fixed number of time steps. Following this, the agent sequentially moves on to the next object in the goal image that is not solved and repeats this process until either all goals are solved or the agent runs out of evaluation time steps. For our purpose, we define a goal as solved when the $\mathbf{z}^{\text{where}}$ component of the best matching object from the observation is closer to the $\mathbf{z}^{\text{where}}$ component of the sub-goal than some threshold.

---

**Algorithm 2** SMORL: Self-Supervised Multi-object RL (Training with Details)

---

**Require:** SCALOR encoder $q_\phi$, goal-conditioned policy $\pi_\theta$, goal-conditioned value function $Q_w$, number of data points from random policy $N$, number of training episodes $K$, number of time steps in the episode $H$
1: Collect $\mathcal{D} = \{\mathbf{o}_i\}_{i=1}^N$ using random initial policy.
2: Train SCALOR on sequences data uniformly sampled from $\mathcal{D}$ using loss described in Eq. 4.
3: Fit prior $p(\mathbf{z}^{\text{where}} \mid \mathbf{z}^{\text{what}})$ to the latent encodings of observations $\{(\mathbf{z}_i^{\text{where}}, \mathbf{z}_i^{\text{what}})\}_{i=1}^N$ obtained using $q_\phi(\mathbf{z}_t \mid \mathbf{z}_{<t}, \mathbf{o}_{\leq t})$.
4: **for** $n = 1, ..., K$ episodes **do**
5:     **for** $t = 1, ..., H$ steps **do**
6:         **if** $t = 1$ **then**
7:             Generate goal $\mathbf{z}_g = (\hat{\mathbf{z}}_g^{\text{where}}, \mathbf{z}_g^{\text{what}})$ using SCALOR and initial observation $\mathbf{o}_1$ (pick random detected object $k$ and substitute $\mathbf{z}^{\text{where}}$ by sampled from prior $\hat{\mathbf{z}}_g^{\text{where}} \sim p(\mathbf{z}^{\text{where}} \mid \mathbf{z}^{\text{what}})$).
8:         **end if**
9:         Encode $\mathbf{z}_t$ using $q_\phi(\mathbf{z}_t \mid \mathbf{z}_{<t}, \mathbf{o}_{\leq t})$.
10:        Get action $\mathbf{a}_t \sim \pi_\theta(\mathbf{a}_t \mid \mathbf{z}_t, \mathbf{z}_g)$.
11:        Execute $\mathbf{a}_t$ and get next state observation $\mathbf{o}_{t+1}$ from environment.
12:        Encode $\mathbf{z}_{t+1}$ using $q_\phi(\mathbf{z}_{t+1} \mid \mathbf{z}_{\leq t}, \mathbf{o}_{\leq t+1})$.
13:        Store $(\mathbf{z}_t, \mathbf{a}_t, \mathbf{z}_{t+1}, \mathbf{z}_g)$ into replay buffer $\mathcal{R}$.
14:        With probability 0.5, replace $\hat{\mathbf{z}}_g^{\text{where}}$ with a sample $p(\mathbf{z}^{\text{where}} \mid \mathbf{z}^{\text{what}})$.    ▷ *Sample "imagined" goals*
15:        Sample transition $(\mathbf{z}, \mathbf{a}, \mathbf{z}', \mathbf{z}_g) \sim \mathcal{R}$.
16:        Compute matching reward signal $R = r(\mathbf{z}', \mathbf{z}_g)$ using Eq. 7.
17:        Minimize Bellman Error using $(\mathbf{z}, \mathbf{a}, \mathbf{z}', \mathbf{z}_g, R)$.
18:     **end for**
19:     **for** $l = t, ..., H$ steps **do**
20:         Sample future state $\mathbf{o}_{h_i}$ that has matching component in observation representation set $\mathbf{z}_{h_i}$ to the original goal $\mathbf{z}_g$, $l < h_i \leq H - 1$.    ▷ *Sample HER "future" goals*
21:         Store $(\mathbf{z}_l, \mathbf{a}_l, \mathbf{z}_{l+1}, \mathbf{z}_{h_i,k})$ into $\mathcal{R}$ (for $k$ such that $\mathbf{z}_{h_i,k}$ is matching the original goal $\mathbf{z}_g$).
22:     **end for**
23: **end for**

---

## D.3 SCALOR

We are using the SCALOR implementation from the original authors[1]. The parameters that were modified from the default settings can be found in Table 1. In particular, we are using $\mathbf{z}^{\text{what}}$ dimension equal to 8 for 1 object and equal to 4 for two objects. We observed that using smaller dimensionalities for $\mathbf{z}^{\text{what}}$ makes the training more stable, if it is possible to train SCALOR with it. As for our purpose, the background model is not important and our environments have stable background, for this work, we are modeling the background with small $\mathbf{z}_{bg} = 1$.

---

[1] https://github.com/JindongJiang/SCALOR

**Algorithm 3** SMORL (Evaluation)

---

**Require:** Trained SMORL agent $\pi_\theta$, goal image $\mathbf{o}_g$, SCALOR encoder $q_\phi$, evaluation episode length $L$, sub-goal episode length $l$
1: Get goal representation $\mathbf{z}_g = \{\mathbf{z}_m\}_{m=1}^N = q_\phi(\mathbf{o}_g)$ where $N$ is the number of recognized objects.
2: Get the number of attempts $K = \frac{L}{l}$.
3: Initialize goal index $m = 1$.
4: Initialize evaluation step $t = 1$.
5: **for** $k = 1, ..., K$ steps **do**
6:     Obtain initial observation $\mathbf{o}_1$ and pick sub-goal $\mathbf{z}_m$.
7:     **for** $s = 1, ..., l$ steps **do**
8:         Encode $\mathbf{z}_t$ using $q_\phi(\mathbf{z}_t \mid \mathbf{z}_{<t}, \mathbf{o}_{\leq t})$.
9:         Get action $\mathbf{a}_t \sim \pi_\theta(\mathbf{a}_t \mid \mathbf{z}_t, \mathbf{z}_m)$.
10:         Execute $\mathbf{a}_t$ and get next observation $\mathbf{o}_{t+1}$ from environment.
11:         Set $t = t + 1$.
12:     **end for**
13:     **if** all sub-goals $\mathbf{z}_m$ are solved **then**
14:         Stop evaluation.
15:     **end if**
16:     Set $m = (m + 1) \bmod N$.
17:     **while** $\mathbf{z}_m$ is solved **do**
18:         Set $m = (m + 1) \bmod N$.
19:     **end while**
20: **end for**

---

During RL training, we process the first observation $o_1$ of each episode 5 times with SCALOR which we found to stabilize the inferred representations. During evaluation, we do the same with the goal images $o_g$ given from the environment.

| Hyper-parameter | Value |
|---|---|
| Optimizer | Adam (Kingma & Ba, 2015) with default settings |
| Number of iterations | 5000 |
| Learning rate | 0.0001 |
| Batch size | 11 |
| Explained Ratio Threshold | 0.1 |
| Number of training points | 10000 |
| Number of cells | 4 |
| Size bias | 0.22 |
| Size variance | 0.12 |
| Ratio bias | 1.0 |
| Ratio variance | 0.3 |

Table 1: SCALOR hyper-parameters.

## D.4 SMORL

We refer to Table 2 for general hyper-parameters of SMORL and to Table 3 for environment specific hyper-parameters of SMORL.

## D.5 PRIOR WORK

For the baselines, i.e. SAC, RIG and Skew-Fit, we started from standard settings and made environment-specific tweaks to tune them for best performance. In particular, significant hyper-parameter search effort (>500 runs) was spent on finding the best SAC parameters for *Multi-Object Visual Rearrange* 2, 3, and 4 objects.

| Hyper-parameter | Value |
|---|---|
| Optimizer | Adam with default settings |
| Exploration Noise | None (SAC policy is stochastic) |
| RL Batch Size | 2048 |
| Reward Scaling | 1 |
| Automatic SAC entropy tuning | yes |
| SAC Soft Update Rate | 0.05 |
| # Training Batches per Time Step | 1 |
| Hidden Activation | ReLU |
| Network Initialization | Xavier uniform |
| Separate Attention for Policy & Q-Function | yes |
| Replay Buffer Size | 100000 |
| Relabeling Fractions Rollout/Future/Imagined Goals | 0.1 / 0.4 / 0.5 |
| Number of Initial Random Samples | 10000 |

Table 2: General hyper-parameters used by SMORL for visual environments.

## E PROBLEMS WITH SCALOR TRACKING DURING RL TRAINING

During our experimentation with the reward specification, we first considered SCALOR's internal tracking of objects. SCALOR assigns each discovered object an ID, and these IDs are in principle propagated over time steps. By matching IDs, one can easily compute distances to the goal $\mathbf{z}_g$ in the space of the $\mathbf{z}^{\text{where}}$ component (because we pick the episode goal from the objects discovered in the first observation during RL training). However, with such a reward specification, the agent was easily finding ways to exploit the biases towards a position in the propagation of the representation to the next time step.

| Hyper-parameter | Push, 1 Obj. | Push, 2 Obj. |
|---|---|---|
| Training Path Length | 15 | 15 |
| Evaluation Path Length | 45 | 75 |
| Learning Rate | 0.001 | 0.0007 |
| Discount Factor | 0.925 | 0.95 |
| Matching Threshold $\alpha$ | 1.2 | 1.3 |
| No Match Reward $r_{\text{no goal}}$ | 0.75 | 1.0 |
| $\mathbf{z}^{\text{what}}$ Dim | 8 | 4 |
| Embedding Dim $d_e$ | 48 | 32 |
| Number of Cond./Uncond. Heads | 3/0 | 1/1 |
| Number of Input-Independent Queries | 0 | 3 |
| Policy Hidden Sizes | $[128, 128]$ | $[128, 128, 128]$ |
| Q-Function Hidden Sizes | $[256, 256, 256]$ | $[128, 128, 128]$ |

| Hyper-parameter | Rearrange, 1 Obj. | Rearrange, 2 Obj. |
|---|---|---|
| Training Path Length | 20 | 20 |
| Evaluation Path Length | 60 | 100 |
| Learning Rate | 0.001 | 0.0005 |
| Discount Factor | 0.95 | 0.925 |
| Matching Threshold $\alpha$ | 1.2 | 1.3 |
| No Match Reward $r_{\text{no goal}}$ | 0.75 | 1.5 |
| $\mathbf{z}^{\text{what}}$ Dim | 8 | 4 |
| Embedding Dim $d_e$ | 48 | 32 |
| Number of Cond./Uncond. Heads | 3/0 | 1/1 |
| Number of Input-Independent Queries | 0 | 3 |
| Policy Hidden Sizes | $[64, 64]$ | $[128, 128, 128]$ |
| Q-Function Hidden Sizes | $[128, 128, 128]$ | $[128, 128, 128]$ |

Table 3: Environment specific hyper-parameters used by SMORL for visual environments.

In particular, one underlying assumption of SCALOR is that "two objects cannot coexist in the same position" (Jiang et al., 2019). However, due to 2D-projecting the 3D objects and possible occlusions, this assumption is not always fulfilled and the RL agent was able to exploit this during training. For example, the agent learned to position the robotic arm exactly above the object, and due to the positional propagation, this object's component was then propagated to the arm. After this, the agent was able to "manipulate" this component just by positioning his arm to the object's goal. This shows the importance of evaluating learned representations in downstream tasks.

