# OpenReview forum: "Self-supervised Visual Reinforcement Learning with Object-centric Representations"
_ICLR.cc/2021/Conference — ICLR 2021 Spotlight_

### Official Review · AnonReviewer4 · 2020-10-25
**Object centric learning using SCALOR in goal-conditioned, model-free RL**

**Rating:** 8
**Confidence:** 3

**Review:**

Summary:

The paper combines an existing generative world model (SCALOR, Jiang et al. 2019) with goal-conditioned attention policy. The method is evaluated on object manipulation environments based on MuJoCo (Todorov et al., 2012), Multiworld (Nair et al. 2018) and a Sawyer arm. The paper is clearly written; the authors discuss challenges and motivate their design choices well throughout the paper.

Score justification:

The paper is mostly incremental but it provides enough contributions for acceptance. SMORL (Algorithm 1; the proposed method) is well-defined and motivated. The method outperforms a strong baseline (Soft Actor-Critic with Hindsight Experience Replay) in an object manipulation task with available ground-truth state representations. In the visual rearranging task, the proposed method performs better than existing self-supervised RL algorithms: RIG (Nair et al., 2019) and Skew-Fit (Pong et al., 2020) when using 1 and 2 objects. Experiments with a larger set of objects which demonstrate the compositional benefit of SMORL would strengthen the paper. I would also be interested in a visualization of the latent object representations learned by SMORL.

Pros:

The paper contributes to improving scene decomposition and object representation learning in model-free RL which has practical applications in robotics and object-oriented RL.

There is a discussion of existing limitations and challenges (limitations of VAEs in visual RL, defining reward functions in goal-conditioned RL), and how SMORL is meant to address them (goal-conditioned attention policy to handle set inputs, incorporating goal and object representations in the reward).

Experiments in the multi-object environments showing that SMORL might be promising with and without ground-truth representations.

Cons:

There is no discussion of the computation cost of SMORL in comparison to the baselines (SAC with HER, RIG, Skew-Fit).

The “compositional” aspect is unclear in the Experiments section. How does the “compositional generative world model” translate to productivity, substitivity or other forms of compositional generalization with respect to the objects in the image?

Experiments with a larger set of objects would help in highlighting the advantage of SMORL in a general multi-object visual RL setting.

Questions during rebuttal period:

Please address and clarify the cons above

Typos and structure:

The link in the abstract leads to an empty page

Equation 1: G (distribution over the set of goals) is once bolded, once in italic

It seems there are two different citation styles used in the paper: (Jiang et al., 2019) and Jiang et al. (2019)

Algorithm 1, line 1: typo “sequences data”

Section header Experiments can be moved to the next page

Typos: “our method scale challenging tasks”, “out code”, “objects identities”, “2 object”, “3 object”, “4 object”

---

> ### Author Response · Authors · 2020-11-19
> **Response to Reviewer #4**
>
> Dear reviewer, thank you for your positive evaluation of our work and for the constructive feedback.
>
> “I would also be interested in a visualization of the latent object representations learned by SMORL.”
>
> We have implemented different analyses of SCALOR’s representation, which are presented in the Appendix A of the updated paper, in particular concerning your questions:
>  - A clustering of $z^\mathrm{what}$ components reveals a clear separation of the underlying objects, see A.1.
>  - We have added some trajectory traversals in A.3.
>
> “There is no discussion of the computation cost of SMORL in comparison to the baselines (SAC with HER, RIG, Skew-Fit).”
>
> Thanks a lot for pointing this out. We have added this discussion into section 5.2 of the paper.
> The main computational cost is in representation learning and the encoding of images to representations during the data collection for RL. Here, we can provide relative comparisons. While SCALOR training takes longer than VAE training because of SCALOR utilizing recurrence over the sequence of observations, it can be done once and then used throughout RL training. Thus, RL training for RIG and VAE is comparable as they also use their trained encoders for inference only. Skew-Fit training on the other side takes much more time as it requires additional VAE re-training during the RL phase.
>
> *“The “compositional” aspect is unclear in the Experiments section. How does the “compositional generative world model” translate to productivity, substitivity or other forms of compositional generalization with respect to the objects in the image?”*
>
> Our architecture creates a factorized representation by design. Also, position and appearance are additionally decoupled. The experiments shown in Fig. 3 with 3 and 4 objects show empirical evidence for its effectivity. The number of different goal configurations grows exponentially with the number of objects. Thus we address a form of compositionality where the tasks are *composed* of largely independent subtasks (namely moving one object at a time). To clarify our scope we reworded the passages addressing the compositionally.  We also show the generalization capabilities of our system on unseen tasks with a different number of objects at test time (see 5.3 of updated paper).
>
> *"Experiments with a larger set of objects would help in highlighting the advantage of SMORL in a general multi-object visual RL setting."*
>
> Unfortunately, we do not have enough computation resources to run all the hyperparameter optimization etc during the rebuttal period for more objects. The experiment reported in Fig. 3 with 3 and 4 objects show that good representations will transfer to good performance using our policy architecture. However, we agree that this is an important future direction.

---

### Official Review · AnonReviewer2 · 2020-10-28
**Official Blind Review #2**

**Rating:** 9
**Confidence:** 5

**Review:**

This paper combines object-centric representations and self-supervised HER goal-conditioned policy learning to learn efficient RL policies for a robot manipulation task.

They use SCALOR as an object-based state representation, and use it to propose semantically meaningful goals for a SAC policy to achieve. They can then leverage this on new tasks to solve them efficiently..

Overall, I found this paper very interesting, clearly written, well executed (with very sensible decisions throughout) and presenting several good ideas, especially in how to leverage the additional object structure effectively. It demonstrates good early results in the novel field of Object-oriented RL.

I have a few comments/questions:

1. The explanation of how the goal-conditioned policies were trained was very clear, and I especially like how you use z_what and z_where to construct novel meaningful goals (which will tend to just force the policy to move objects around, but that is a good prior for your environment!). However how the “evaluation on a novel task” is done wasn’t as clear (i.e. when we try to implement a given goal or sequence of goals). More precisely, it is said in several places that the goal z_g is decomposed in sub-tasks where only one of the slots is used as the target.
   1. Could you provide more details on exactly how that is done? Do you learn p(z^where) on task later?
   2. What happens when some of the “objects” aren’t achievable or controllable? E.g. I’d expect that you see a slot which represents the robot arm, is this treated differently?
2. How good are SCALOR representations in your environment?
   1. It would be very helpful to show samples / traversals in the Appendix.
   2. Similarly, comparing to the GT information you provide would be interesting (e.g. try to decode it? I understand you’d have to match the slots up unfortunately)
   3. Did you try continuing training SCALOR in the RL phase? It would make the results stronger and less reliant on a good random exploration strategy.
3. Did the hard matching cause issues while learning? I’d guess that the argmin is not too problematic because it is used in the reward computation only, but if you’d consider extending this setting to learning the goal proposal function z_g, this seems like a limitation?
   1. The explanation about the issue of using tracking as part of the model directly wasn’t especially clear to me. It might deserve a bit more expansion, especially in the Appendix?
4. How complex are the observations of the environment?
   1. Could you add more samples of the environment’s observations?
   2. It seems like the environment chosen is extremely similar to the standard Gym Fetch environment, did you try using it instead? https://gym.openai.com/envs/FetchPickAndPlace-v0/
5. It is not entirely true that MONet/IODINE “do not contain disentangled and interpretable features like position and scale”.
It is true that they are not explicitly enforced (like done in SCALOR), but they do arise quite easily purely unsupervised.
Especially, in my experience with both of these models, obtaining (and identifying) “position” latents is rather easy. See for example Figure 5 in [1] and this animation [2].

So in summary, I believe this is a strong paper in a budding field, which deserves publication at ICLR and may interest many people there.

* [1] https://arxiv.org/abs/1901.11390
* [2] https://twitter.com/cpburgess_/status/1091220207941701632

---

> ### Author Response · Authors · 2020-11-19
> **Response to Reviewer #2**
>
> Dear reviewer, thanks a lot for your review. We are glad that you like our work.
>
> *“How the “evaluation on a novel task” is done wasn’t as clear (i.e. when we try to implement a given goal or sequence of goals)”*
>
> The algorithm sequentially attempts to solve all the recognized sub-goals with different $z^{\mathrm{where}}$. We clarified this in the paper, also described precisely in Algorithm 3 in the Appendix. We believe that using a more sophisticated planning algorithm would allow for solving more complex tasks during evaluation. We discuss this in the paper now (Conclusions).
>
> *“What happens when some of the “objects” aren’t achievable or controllable?.”*
>
> Right now, the agent would still attempt to “solve” these tasks. At test time, the agent has several attempts to solve sub-goals with a limited time-budget, so it would not get stuck on these tasks. A discussion and future work direction is added to the Conclusion.
>
> *“E.g. you see a slot which represents the robot arm, is this treated differently?”*
>
> The arm is typically represented by its own slot. It is not treated differently, but just considered as one of the discovered objects that are controllable. This way, one of the discovered tasks is actually reaching (i.e. manipulating yourself) of the robot arm to a specific location. We have verified that robots do quickly learn this sub-task also and can include this in the Appendix if needed.
>
> *“How good are SCALOR representations in your environment?”*
>
> This is a really good question. We have implemented different analyses of SCALOR’s representation, which are presented in the Appendix A of the updated paper. To summarize:
> A clustering of $z^\mathrm{what}$ components reveals a clear separation of the underlying objects, see A.1.
> The representation of SCALOR is highly disentangled according to the Mutual Information Gap (MIG) measure in contrast to VAE representations, see A.2. As you have observed correctly, this requires a matching which we do using the clustering above.
>
> *“Did you try continuing training SCALOR in the RL phase? It would make the results stronger and less reliant on a good random exploration strategy.”*
>
> For this work, we restricted ourselves to training SCALOR on data from a random policy. As online training of SCALOR would take much more computational resources and potentially would be less stable (this intuition is supported by comparing the online Skew-Fit architecture with the passive RIG method). Nevertheless, it is an interesting future work direction (see second part of conclusion in the updated paper).
>
> *“The explanation about the issue of using tracking as part of the model directly wasn’t especially clear to me. It might deserve a bit more expansion, especially in the Appendix?”*
>
> We provide more detail in Appendix E. The imperfection of the tracking algorithm provided by SCALOR can be exploited by the RL agent.
>
> *“Could you add more samples of the environment’s observations?”*
>
> We have added some trajectory traversals in A.3.
>
> *“It seems like the environment chosen is extremely similar to the standard Gym Fetch environment, did you try using it instead?”*
>
> The reason for choosing the multiworld environments is because it was used in the literature on visual RL. In principle the Gym Fetch environment could be modified to provide visual observations and multiple objects.
>
> *“It is not entirely true that MONet/IODINE “do not contain disentangled and interpretable features like position and scale”.
> It is true that they are not explicitly enforced (like done in SCALOR), but they do arise quite easily purely unsupervised. ”*
>
> Thanks for spotting this. We have clarified this point in the updated paper.

---

### Official Review · AnonReviewer1 · 2020-10-28

**Rating:** 7
**Confidence:** 5

**Review:**

This work proposes to use object-centric unsupervised representation learning for self-supervised goal-conditioned RL, as opposed to prior work that assumes no particular structure on the learned representations (eg. VAEs). The proposed method, self-supervised multi-object RL (SMORL), uses the SCALOR architecture from prior work, then modifies the policy representation with single-object attention and also the reward function in RL with imagined goals (RIG). The results show that the method can learn simulated pushing and rearranging tasks in a self-supervised way with up to 4 objects in the scene, and outperforms RIG and Skew-Fit on pushing tasks. The proposed method is sufficiently novel, explores an important direction for self-supervised learning, and the results are quite strong.

The main motivation argued is that in more complex environments, it is difficult for fully unstructured reconstruction-based representation learning methods such as VAEs to recover a disentangled representation. This then causes difficulties running self-supervised RL algorithms such as RIG, which use the representation to compress the input, to set meaningful exploration goals, and for evaluating the reward. Using object-centric representations makes the representation more disentangled and improves RL, as demonstrated by the results. The paper would be strengthened by direct analysis of this hypothesis: for instance, correlations or measures of disentanglement between ground truth state and learned representation for SCALOR vs. VAE in the environments tested, particularly as number of objects increase.

The key novel contributions lie in how SCALOR is integrated with self-supervised learning. First, after learning the SCALOR representation from data, the proposed policy uses an attention mechanism to pay attention to reaching the goal for a single object at a time. One detail I did not understand was how the policy operates at test time. At training time, exactly one z^where is changed and the policy attempts to match that object. At test time, potentially many z^where could be different, so do you cycle between all the objects? Also, the paper would benefit from an ablation where you explore the choice of attention architecture; you could imagine that simply learning the object-centric representation and treating it as a flat representation like a VAE could also provide gains, so it is important to disentangle that effect from the novel policy architecture. The policy contribution is evaluated in Figure 3, where the success of SMORL+GT shows that the architecture makes manipulating a large number of objects possible.

The other differences to RIG have to do with goals and rewards. During self-supervised training, goals are sampled by sampling a new z^where for a single object, encouraging manipulation of exactly one object. This proposal seems logical, although in the long run it could potentially be an assumption that would not scale beyond object repositioning tasks. The reward is also modified to use the SCALOR latent, to penalize distance to the closest z^what object as the current goal with a threshold alpha for detecting the matching object. Again, the proposed reward function would be better evaluated with an ablation where you use the original reward from prior work -||z - z_g||.

The experiments show that the proposed method SMORL outperforms RIG and Skew-Fit on visual pushing tasks with many objects (and “rearranging”, which is pushing with random initial positions of objects - having both sets of experiments potentially seems a bit redundant since it seems rearranging is strictly more difficult). SMORL is worse than an oracle (SAC+GT) which uses ground truth state information, but it seems to tend towards the oracle performance on even the more difficult tasks (seeing longer versions of the learning curve that show asymptotic performance would help understand this better).

Generally, the results on multi-object manipulation and self-supervised learning are strong. Further experiments as mentioned above would better allow the contributions to understood independently.

Minor comments

The related work is a bit thin and narrow - it addresses the nearest-neighbor works well but does not address self-supervised methods and robotics methods more broadly; it would be best to use the related work to make the paper more understandable to the broader community who are not embedded in goal-conditioned RL (and potentially put it in at section 2 instead of at the end).

Page 6: “Out code” -> “our code”

“In general, we expect that it is beneficial for the policy to not always attend to entities conditional on the goal; we thus allow some heads to only attend to additional learned parametric queries (left out above for notational clarity).” - did not understand this, it would be good to explain further what type of information this would include, and perhaps describe formally in the appendix.

---

> ### Author Response · Authors · 2020-11-19
> **Response to Reviewer #1**
>
> Dear reviewer, thank you for your positive evaluation of our work and for the constructive feedback.
>
> *“The paper would be strengthened by direct analysis of this hypothesis: for instance, correlations or measures of disentanglement between ground truth state and learned representation for SCALOR vs. VAE in the environments tested”*
>
> This is a really good suggestion, which we implemented now. We have added several parts in appendix that are addressing SCALOR’s representation properties. In particular, we have added a disentanglement analysis in App A.2, where we compute the mutual information gap (MIG) for VAE and SCALOR representations. As the VAE’s MIG scores were low for 1 and 2 objects, we decided to add only one comparison, however, if needed, other comparisons could be also added. In addition, we have added a mutual information matrix to show a more detailed picture of the disentanglement of SCALOR components.
>
> *“At test time, potentially many $z^\mathrm{where}$ could be different, so do you cycle between all the objects? ”*
>
> Yes, we sequentially try to solve all the recognized sub-goals with different $z^\mathrm{where}$. We clarified this in the paper, also described precisely in Algorithm 3 in the Appendix.
> We discuss this aspect in the conclusion now (see second part of conclusion in the updated paper).  We believe that using a more sophisticated planning algorithm would allow for solving more complex tasks during evaluation.
>
> *“The paper would benefit from an ablation where you explore the choice of attention architecture; you could imagine that simply learning the object-centric representation and treating it as a flat representation like a VAE could also provide gains”*
>
> The choice of our policy was motivated by two different properties:
>  - SCALOR representation are unordered, potentially different-sized sets
>  - The policy should be able to concentrate on parts of the representation set
>
> Due to the first property we can not directly convert SCALOR representations to a fixed-length vector (the number of recognised objects can change, e.g. because of occlusions). We have done an ablation study where we explored 3 different choices of policies compatible with SCALOR representations. The results are presented in Appendix B of the updated paper.  First, we checked the importance of goal-conditional heads and an unconditional head in our architecture. We find out that without the goal-conditional heads, the SMORL algorithm performs significantly worse (while both are contributing to the final performance), showing the importance of goal-conditioned attention.  Next, we substitute the goal-conditioned attention module with the Set aggregation algorithm called DeepSet [1]. We observe that SMORL with DeepSets can also perform competitively on the two objects tasks, however, it is significantly worse on the out-of-distribution (OoD) task with one object. This shows that goal-condition attention contributes to the OoD generalization properties of our architecture.
>
> *“The proposed reward function would be better evaluated with an ablation where you use the original reward from prior work $-||z - z_g||$.”*
>
> Indeed, such an ablation would be interesting, however given that representations are unordered it would require an additional matching of the recognized objects of current input set $z$ and the goal set $z_g$, so it cannot be performed.
>
> *“Seeing longer versions of the learning curve that show asymptotic performance would help understand this better”*
>
> Thanks for mentioning this. We now have trained both SMORL and RIG twice longer and indeed get improved SMORL performance with average distance to the goal = 0.14 (2 objects), see plots in App. C. As the performance curve of SMORL still suggests further improvements with even more training, we will train longer and update the plots accordingly (not feasible in the rebuttal period due to computational restrictions).
>
> *“The related work is a bit thin and narrow - it addresses the nearest-neighbor works well but does not address self-supervised methods and robotics methods more broadly”*
>
> We decided to add an introductory passage where we cover the importance of self-supervised methods and visual RL. There, we also covered how self-supervised methods are used in goal-based RL as this could be less known. We also moved the related work section after the introduction, as per your suggestion.
>
> *"It would be good to explain further what type of information this would include, and perhaps describe formally in the appendix."*
>
> Thanks for spotting this. We added a detailed description of the unconditional goals in Appendix D (D.1 of the updated paper).
>
> [1]: Zaheer, M. et al. Deep Sets. Advances in Neural Information Processing Systems 30, 3391–3401 (2017).

---

> > ### Comment · AnonReviewer1 · 2020-11-20
> > **Update**
> >
> > Thank you for running the additional experiments, and I understand why the two initially proposed ablations are infeasible. The representation learning visualizations and ablations helped to understand how SMORL is working, and my concerns have been addressed.
> >
> > Related work is definitely better now, but still quite terse. There is more work that could be covered in self-supervised RL (eg. [1, 2]).
> > [1] Unsupervised control through non-parametric discriminative rewards. Wade-Farley et al. 2018.
> > [2] Active learning of inverse models with intrinsically motivated goal exploration in robots. Baranes et al. 2013.
> >
> > In light of the revisions I have raised my score.

---

> > > ### Author Response · Authors · 2020-11-25
> > > **Added references**
> > >
> > > Thank you for pointing out those references. We included them into our related work section.

---

### Official Review · AnonReviewer3 · 2020-10-28
**Interesting combination of object-centric representations and RL but insufficient experimental results**

**Rating:** 5
**Confidence:** 5

**Review:**

### Summary
The paper proposes to use object-centric representations for RL, which can efficiently handle multiple objects in the scene. To learn a policy that can take a variable number of object observations, the paper proposes the goal-conditioned attention policy, which can focus on objects of interests to achieve each sub-goal, and thus reduce the combinatorial complexity of multiple objects. The goal-conditioned attention policy can be efficiently trained with hindsight experience replay on the object-centric goal representations. The experiments demonstrate the superior performance of the goal-conditioned attention policy on dealing with multiple objects.

### Strengths
- The idea of learning composable object-centric visual representations and goal-conditioned attention policy is an intuitive and plausible way to tackle combinatorial challenges in multi-object manipulation tasks.
- The experiments with ground truth states show that the proposed goal-conditioned attention policy can effectively handle multiple object manipulation tasks.

### Weaknesses

- Based on Figure 4, any of the methods without ground truth states solve the tasks. Although the proposed method shows better learning performance in the Visual Rearranging task, the improvement is marginal to claim the proposed method can solve the tasks.
- Why only up to 2 objects are considered in Figure 4? The proposed method has the advantage of dealing with multiple objects but did not show the benefit. It would be more convincing if the proposed method can reasonably deal with more than 2 objects.
- The baseline could include recent visual policy learning methods using data augmentation [1,2].
- The paper claims "Self-supervised RL" but it is not clear which part of the method is trained with self-supervised learning. The proposed method seems to consist of unsupervised representation learning and reinforcement learning.
- The proposed method assumes that the sub-goals are independent of each other but it is not true in many cases, e.g., collisions between objects.
- One of the claims in the paper is that the proposed representations and policy can work with a variable number of objects but the experiments do not cover this setup.

### Questions and additional feedback
- It would be better to include the architecture of the visual encoders for baselines and the proposed method.
- It could be good to show the quality of learned representations, such as object pose prediction and classification.
- The training time might be too short. The proposed method can be trained more (e.g., 1e6 environment steps).
- The website link is provided but nothing is there.
- How does the policy decide when to switch to the next sub-goal? Including how to rollout an episode toward sequential sub-goals with the proposed model would be helpful.

### Overall assessment
The proposed method is intuitive and tackles an important problem of multi-object manipulation. However, the experiments and results are not yet convincing to claim the advantage in dealing with multiple objects. Overall, the reviewer thinks the paper requires more thorough experiments and is not ready to be published.


[1] Laskin et al., Reinforcement Learning with Augmented Data

[2] Kostrikov et al., Image Augmentation Is All You Need: Regularizing Deep Reinforcement Learning from Pixels

---

> ### Author Response · Authors · 2020-11-19
> **Response to Reviewer #3 Part 1/2**
>
> Dear Reviewer 3, thank you for your review.
>
> *"The paper claims "Self-supervised RL" but it is not clear which part of the method is trained with self-supervised learning. The proposed method seems to consist of unsupervised representation learning and reinforcement learning."*
>
> In our setting, both goals and rewards are constructed intrinsically from observations, and in this sense our method is self-supervised. Generally, the term self-supervised RL refers to methods that acquire a diverse repertoire of general-purpose robotic skills without reward signals using only observations. These skills can be reused and combined during test time. This view seems consistent with the literature we examined. To clarify this we added an introductory passage to the related work where we covered how self-supervision is used in goal-based RL without external rewards.
>
> *"The proposed method assumes that the sub-goals are independent of each other but it is not true in many cases, e.g., collisions between objects."*
>
> In the tasks we are considering, sub-goals are mostly independent as it is possible to achieve each of them independently without influencing other sub-goals. However, we agree that going towards more complex tasks such as object stacking would require rethinking this assumption, and is very interesting for future work. Therefore, we now mention this as a potential future direction in Sec. 6.
>
>
> *"One of the claims in the paper is that the proposed representations and policy can work with a variable number of objects but the experiments do not cover this setup."*
>
> We have added an additional experiment (see 5.3) where we evaluate our policy trained on 2 objects on 1 objects environment, showing that its performance is comparable to a policy trained on only one object.
>
>
> *"It could be good to show the quality of learned representations, such as object pose prediction and classification."*
>
> We have implemented different analyses of SCALOR’s representation, which are presented in the Appendix A of the updated paper. To summarize:
>
>  - A clustering of $z^\mathrm{what}$ components reveals a clear separation of the underlying objects (see A.1), which indicates that the representations can easily be used for tasks such as classification.
>  - The representation of SCALOR is highly disentangled according to the Mutual Information Gap (MIG) measure in contrast to VAE representations, see A.2.
>
>
> *"How does the policy decide when to switch to the next sub-goal?"*
>
> The algorithm sequentially attempts to solve all the recognized sub-goals. We now clarify this in the paper and also added Algorithm 3 in the Appendix that describes it precisely. We also discuss other potential approaches to implement this in Section 6 now.
>
> *"The training time might be too short. The proposed method can be trained more (e.g., 1e6 environment steps)."*
>
> We now have trained both SMORL and RIG twice longer and indeed get improved SMORL performance with average distance to the goal = 0.14 (2 objects), see plots in Appendix C. We will replace the plots in the main paper when all curves are finished.
>
> *"Although the proposed method shows better learning performance in the Visual Rearranging task, the improvement is marginal to claim the proposed method can solve the tasks."*
>
>  As we report above, longer training shows improvements upon the initially reported results. Nevertheless, it also alerts that progress is needed on the representation side, to close the gap to the ground truth curves. Regarding this gap, there is an inherent problem: the reward for training is defined on the camera image, which is actually having a slant view on the scene (as used in [1]). Thus the measure that is optimized by the agent is not directly matching our evaluation criterion.
>
> [1]: Nair, Ashvin V., et al. "Visual reinforcement learning with imagined goals." Advances in Neural Information Processing Systems. 2018.

---

> > ### Author Response · Authors · 2020-11-19
> > **Response to Reviewer #3 Part 2/2**
> >
> > *“The baseline could include recent visual policy learning methods using data augmentation [1,2].”*
> >
> > Thanks for this suggestion. However, we believe these methods are not applicable for our setting because 1) they are designed for a single-task setting, whereas we are targeting a multi-task setting, 2) these methods rely on a reward signal being provided to construct augmented data, whereas we have no external supervision available. To our knowledge, these methods have not yet been applied to the setting we are considering, and therefore it is not clear to us how to use them as baselines.
> >
> > *“It would be more convincing if the proposed method can reasonably deal with more than 2 objects.”*
> >
> > Unfortunately, we do not have enough computation resources to run all the hyperparameter optimization, etc during the rebuttal period for more objects. The experiment reported in Fig.3. with 3 and 4 objects shows that good representations will transfer to good performance using our policy architecture. However, we agree that this is an important future direction.
> >
> > *“The website link is provided but nothing is there.”*
> >
> > We are sorry for delaying the website upload. We have updated the website with visualization of SMORL trained on GT representations and SCALOR representations. Code will follow.

---

> > > ### Comment · AnonReviewer3 · 2020-11-20
> > > **Thank you for your detailed response**
> > >
> > > Thank you for your detailed response. The response resolves most of my questions and concerns, and thus increase my rating from 4 to 5.
> > >
> > > However, the reviewer considers the scalability of the proposed goal-conditioned attention policy as the primary contribution of the paper and this is not empirically shown in the paper. Therefore, the experiments on Visual Push and Visual Rearrangement experiments with 3-4 objects are essential to make this paper convincing.

---

### Author Response · Authors · 2020-11-19
**General Response**

Dear reviewers, we now updated our paper to address your comments and questions. In particular, we added
 - An out-of-distribution generalization experiment (Section 5.3)
 - An analysis of the learned SCALOR representations in terms of clustering (Appendix A.1) and disentanglement (Appendix A.2)
 - Environment traversals and how SCALOR processes them (Appendix A.3)
 - An ablation study about the impact of different choices for attention heads (Appendix B)
- Curves for longer training time for the Visual Rearrange 2 objects experiment (Appendix C)
- An expanded related work section (Section 2) in which we more broadly address the literature around our method

We now also added videos of the learned policies to the project website. We apologize for the delay in doing so. We will address individual questions and concerns as direct answers to the reviews. If you have any further requests or questions that we could address in the remaining rebuttal period, please do not hesitate to comment.

---

### Author Response · Authors · 2020-11-25
**Final Revision**

Dear reviewers, we uploaded a final revision where we cleaned up some minor details and added a few more references.
Thank you for engaging with us during the review period!

---

### Decision · Program_Chairs · 2021-01-07
**Final Decision**

**Decision:**

Accept (Spotlight)

**Comment:**

This paper proposes a self supervised learning algorithm to compute object-centric representations for efficient RL in the context of robot manipulation tasks.

The key idea is to learn an object-centric representation (using prior work on SCALOR) and use this to intrinsically generate goals for a SAC policy to achieve. The policy is a goal-conditioned attention policy. The evaluation metric is a set of tasks to manipulate objects for a visual rearrangement task.

${\bf Pros}: $
1. The baselines are reasonable and consist of other unsupervised RL algorithms in recent literature.

2. Object-oriented RL is a growing area of interest and this paper proposes a reasonably novel and validated set of ideas in this domain. I believe it will be of significant interest and potentially make an impact on research in robotics and deep reinforcement learning.

3. The goal-conditioned attention policy can handle realistic scenarios, namely -- multi-object manipulation tasks

4. The attention mechanism also provides a reasonable solution to mitigate combinatorial hardness in multi-object environments

${\bf Cons}$:

1. Some of the reviewers felt that the experimental results from pixel inputs could have been pushed further. However, since the setup and algorithm is relatively novel, there are already many moving parts and this paper seems like a step in that direction

2. Experiments with larger set of objects would have been interesting to investigate and report.